# Mindful Eating Questionnaire: Validation and Reliability in Romanian Adults

**DOI:** 10.3390/ijerph191710517

**Published:** 2022-08-24

**Authors:** Denis Mihai Serban, Costela Lacrimioara Serban, Sorin Ursoniu, Sandra Putnoky, Radu Dumitru Moleriu, Salomeia Putnoky

**Affiliations:** 1Department of Obstetrics-Gynecology, “Victor Babes” University of Medicine and Pharmacy, 300041 Timisoara, Romania; 2Department of Functional Sciences, Center for Translational Research and Systems Medicine, “Victor Babes” University of Medicine and Pharmacy, 300041 Timisoara, Romania; 3Psychiatric Clinic, Emergency Clinical County Hospital, 300425 Timisoara, Romania; 4Department of Mathematics and Computer Science, West University of Timisoara, 300223 Timisoara, Romania; 5Department of Microbiology, Centre for Studies in Preventive Medicine, “Victor Babes” University of Medicine and Pharmacy, 300041 Timisoara, Romania

**Keywords:** mindful eating, MEQ, validation

## Abstract

Mindful eating may play an important role in long-term weight maintenance. In interventions aiming at weight reduction, increasing the levels of mindful eating was associated with higher levels of success and lower levels of weight rebound in the long run. This study aimed to determine the validity and reliability of a mindful eating questionnaire for Romanian adults using Framson’s Mindful Eating Questionnaire (MEQ). To calculate the internal (n = 495) and external (n = 45) reliability, a general population sample was taken. Construct validity was assessed using the “known groups” method: dietitians (n = 70), sports professionals (n = 52), and individuals with overweight and obesity (n = 200). Convergent validity tested the association between the MEQ score and demographic characteristics of the total sample (n = 617). The internal (0.72) and external (0.83) reliability were adequate. Dietitians and sports professionals had overall lower scores, meaning more mindful eating compared to the group of individuals with overweight and obesity. The lower mindful eating practice was associated with the presence of excess weight, suboptimal health status perception, higher levels of stress and younger age. The Romanian version of the MEQ is a reliable and valid tool for measuring mindfulness of eating in adults.

## 1. Introduction

In Romania, the recognition of obesity by the general population as a health issue is still a problem, while recently, 58.6% of adults and 30.6% of children reported as overweight and obese [1]. Interventions aiming reduction of excess weight are hindered by the late perception of obesity—the tipping point of excess weight being obesity-related stigma or comorbidities—and by communication barriers between patients and health professionals due to conflicting beliefs on excess weight [2]. Weight loss techniques based on diets focusing exclusively on rules of eating are difficult to follow, and their success is limited in the long run [3,4].

Mindfulness is a practice in which the main focus of an individual is to be aware of the present moment, pay attention to the actual experience and notice nonjudgmentally the thoughts, body sensations and emotions that arise in correlation to the environment. Mindfulness of eating is an approach that focuses on savoring the moment and food and encouraging the individual’s full presence for the eating experience. Though its main purpose is not to lose weight and it has less to do with counting calories or rules of food intake, mindful eating was reported to be associated with positive behavioral change and weight loss [5].

The early work of Rodin [6] showed that mindfulness training produced a shift in eating habits by diminishing the influence of external factors, such as the form and smell of food, to increasing the influence of internal factors, such as appetite. The reported outcomes of subsequent interventions aiming to improve the level of mindful eating were healthier food intake and a reduction of energy intake and some loss of excess weight [7,8,9].

Currently, there are no validated tools to measure the mindful eating of the general adult public in Romania. This study aimed to determine the validity and reliability of the Mindful Eating Questionnaire, developed by Framson [10] in the Romanian adult population, to be further used in subsequent research.

## 2. Materials and Methods

The Mindful Eating Questionnaire (MEQ) developed by Framson [10] was used as an assessment tool in several projects focused on obesity and its complications [11,12,13] and was validated in several other languages [14,15,16,17,18].

MEQ includes 28 items grouped into five domains: awareness (7 questions, possible score ranging from 7 to 28), distraction (3 questions, possible score ranging from 3 to 12), disinhibition (8 questions, possible score ranging from 8 to 32), emotional response (4 questions, possible score ranging from 4 to 16) and external cue (4 questions, possible score ranging from 4 to 16) [10].

For our purposes, we have used the original version developed by Framson [10] in the following steps (Figure 1):Step 1. The original MEQ was translated to Romanian and then back to English by two independent translators. The English original MEQ version was compared with the backward translated version, followed by minor corrections in the Romanian version (R-MEQ).Step 2. Pretesting of R-MEQ was performed on 12 volunteers, 5 medical doctors and 6 medical students and a Romanian language specialist, working in a panel. The purpose of pretesting was to review the understanding and improve the original meaning of each question. The outcome was minor improvements in the wording of some questions, and the final version of the questionnaire was issued (Appendix A: Romanian version of the Mindful Eating Questionnaire).Step 3. The R-MEQ was applied to a sample of 495 individuals from the general population who were recruited, mostly in the online environment, between June and July 2022. At this step, the internal consistency of the questionnaire per subscale and per total was determined.Step 4. Of 262 of the participants who provided contact details for further involvement in studies, 70 individuals were randomly selected 3 weeks after the initial answer and invited to answer the questionnaire again. The response rate was 64.3%. Using the answers of 45 responders, the external reliability of the questionnaire was determined. This sample size ensures a power of at least 80%, with a margin of error of 5% [19]. The research hypotheses were that the Romanian version would have acceptable levels of internal (Cronbach’s alpha > 0.7 [20]) and external reliability ICC > 0.5 and weighted kappa > 0.5 [21].Step 5. Construct validity was assessed by sampling 70 specialists in dietetics and 52 sports professionals and a subset of individuals with excess weight from the general population using the “known-groups” method. Our hypothesis was that both groups of professionals, sports and dietetics, would have a more mindful eating approach than the population with excess weight.Step 6. Convergent validity was assessed in the general populational group, including the groups of dietetics and sports professionals, exploring associations between MEQ scores and subscores and social, demographic and anthropometric characteristics. The sample size provided by this component ensures a power of at least 80%, with a margin of error of 5%. Our hypothesis was that MEQ scores would be influenced by gender, age, the presence of excess weight, perceived health status and perceived levels of stress.

Besides specific questions, the survey also included demographic questions such as gender, age, self-reported weight and height, self-evaluation of health status (with five possible ordinal categories from excellent to poor), self-evaluation of the level of perceived stress (ordinal categories from 1 to 10, higher scores associated with higher levels of stress) and marital status (nominal with 6 categories).

### Data Management and Data Analysis

Up to step 3, the paper-based version of the questionnaire was used. For the following steps, an electronic collection tool was used. All questions were required for completing and submitting the questionnaire on the platform; therefore, no missing answers were recorded. To obtain the general population sample, the questionnaire was promoted on social media. Dietitians and sports professionals were invited by research team members in enclosed circles to answer the questionnaire. The response rate was 67.9% in the dietitians’ group and 54.0% in the sports professionals’ group.

After the closure of the questionnaires, the databases were exported, and IMB-SPSS version 21 was used to process the data. Subscale scores (Awareness, Distraction, Disinhibition, Emotional and External) and total scores were calculated as presented by Framson [10]. Each item was scored from 1 to 4. For items 1, 2, 6, 7, 9, 11, 13, 17, 18, 19, 27 and 28, the scoring was reversed. Each subscale score was calculated as the mean of items. The summary score was the mean of the 5 subscales. Lower scores are associated with a more mindful eating approach and higher scores with a less mindful approach.

Body mass index (BMI) was calculated as the fraction of the weight in kilograms divided by the squared height in meters. BMI was classified in 4 categories: underweight (<18.5 kg/m^2^), normal weight (18.5–24.99 kg/m^2^), overweight (25–29.99 kg/m^2^) and obese (≥30 kg/m^2^). Body mass categories were reclassified without excess weight (included categories underweight and normal weight) and with excess weight (included categories overweight and obese). Some of the demographic variables were recoded, such as self-evaluation of health status (excellent and very good versus others), perceived levels of stress were recoded dichotomic (up to 6 points—low stress versus 7 and over—high stress) and marital status was recorded as with a partner (married or in a relationship) versus without a partner. Values of MEQ scores and subscales were transformed into quartiles and used for kappa statistics. The following thresholds were used for interpretation of internal consistency: 0.21–0.39 Minimal, 0.40–0.59 Weak, 0.60–0.79 Moderate, 0.80–0.90 Strong, above 0.90 Almost Perfect. For ICC: below 0.5 indicate poor reliability, between 0.5 and 0.75 moderate reliability, between 0.75 and 0.9 good reliability, and any value above 0.9 indicates excellent reliability; for Weighted Kappa > 0.80—very good agreement, 0.61–0.80 good agreement, 0.41–0.60 moderate agreement, 0.21–0.40 fair agreement and <0.20 poor agreement [20,21,22,23,24].

Continuous data not assuming parametric distribution are presented as medians (interquartile range (IQR)). Categorical data are presented as percentages. Normal distribution was tested with the Kolmogorov–Wilson test. For comparing non-parametric data on a 2-category factor, the Mann–Whitney test was used. For three or more groups with data that did not assume normality, Kruskal–Wallis test was the choice. Bonferroni correction was applied for several comparisons following the significant Kruskal–Wallis test. For correlation, the Spearman correlation was used. The size effect was obtained for the Mann–Whitney test by using the formula r = z/square root of N, where r = size effect, z = z score obtained as part of the test and N = total number of individuals included in the test. The size effect was computed for Spearman’s correlation with the following formula r = (rho)^2^, where r = size effect and rho = Spearman’s rho were obtained as part of the test.

## 3. Results

The feminine gender represented more than 80% of the general population and of the dietitians’ group, and males represented almost 80% of the sports professionals’ group. Over 70% of all participants were in a relationship. Dietitians and sports professionals have higher proportions of optimal health status self-evaluation (over 60% in both groups) compared to the general population group, with only 32.6% of responders in this category. Almost half of the general population perceive high levels of stress, with less than one-third of dietitians and sports professionals declaring this high level of stress. The median age and (IQR) were 32.0 (20.0) years for the general population, 37.0 (16.0) years for the dietitians’ group and 31.5 (13.0) years for the sports professionals’ group. The median BMI and (IQR) were 23.9 (6.7) kg/m^2^ for the general population, 21.3 (4.4) kg/m^2^ for the dietitians’ group and 24.1 (4.4) kg/m^2^ for the sport professionals’ group. Dietitians are older than other groups and have lower BMI as compared to other groups. Demographic variables per study group are shown in Table 1.

Cronbach’s alfa assessed internal reliability on a general population sample (n = 495). Per the entire questionnaire, it had an overall alpha value of 0.72, larger than the threshold of 0.7. For subscales, Cronbach’s alfa was above 0.67, except for the External subscale, which had an alpha value of 0.55 (Table 2).

External consistency or test–retest reliability was assessed in a subgroup (n = 45). The total intraclass correlation coefficient was 0.83. Subscale scores varied from 0.6 for External to 0.88 in Disinhibition (Table 2). Weighted kappa values were all adequate, except for the external subscale, which showed moderate agreement.

### 3.1. Assessment of Construct Validity

For the construct validity, each group of professionals, dieticians and sports specialists, was compared with individuals with overweight and obesity extracted from the populational group and between each other (Table 3). Individuals with overweight and obesity (with a median of 2.2 (0.5)) scored a MEQ total score higher than both dietitians (median of 1.9 (0.6)) and sports professionals (median of 2.1 (0.8)), demoting a lower mindful approach. The same trend was observed in the Distraction, Disinhibition and Emotional subscales. For Awareness, only dietitians scored significantly less. No significant difference was observed between the groups in the External subscale. Per total MEQ score, the size effect of the comparison between individuals with overweight and obesity and dietitians was r = 0.36, accounting for medium effect size and between individuals with overweight and obesity and sports professionals was r = 0.22, accounting for a small to medium effect size.

### 3.2. Assessment of Convergent Validity

Table 4 presents the subscale scores and total scores for a sample of 617 responders by different demographic characteristics of the sample. Overall, gender did not influence the MEQ score. Instead, higher scores, denoting lower levels of mindful eating, were recorded in individuals with excess weight, those with suboptimal health status, and individuals reporting higher levels of stress. Awareness scores are higher in those with excess weight and with perceived suboptimal health status. Distraction scores are higher in those with suboptimal perceived health status, higher levels of stress, and patients with excess weight. Disinhibition scores are higher in males, in those with excess weight, those with suboptimal perceived health status, and higher levels of stress. All measured demographics influenced the levels of the emotional component of mindful eating, which is higher in females, individuals with excess weight, higher levels of perceived stress and lower levels of perceived health status. External cues scores are lower in those with excess weight with high levels of stress. Age is in a significant indirect relationship with the MEQ score, but the effect size (r^2^ = 0.01) is very small. BMI is in a significant direct relationship with the MEQ score, but the effect size (r^2^ = 0.08) is very small.

## 4. Discussion

Since currently there is no validated tool to collect information on mindful eating in Romanian adults, the present study aimed to validate and adapt to the Romanian language the Mindful Eating Questionnaire, developed by Framson et al. [10], so it could be used in subsequent research and clinical practice.

The results from subscales and the overall questionnaire were used to assess the internal reliability, external reliability, construct validity and convergent validity of the questionnaire (Table 2, Table 3 and Table 4). For internal reliability, the original questionnaire reported a Cronbach’s alpha of 0.64, with subscale alpha ranging from 0.64 to 0.83 [10]. Other validations reported similar values [14,15,16,17]. The intraclass correlation coefficients in other validations in different languages ranged from 0.104 to 0.66 [15,16]. For current validation, Cronbach’s alpha per overall questionnaire is above the threshold of 0.7 (Table 2) with moderate internal validity. For the overall questionnaire, ICC and kappa are with good reliability, respective with good agreement. Within the subscales, the internal and external validity is acceptable only for the Disinhibition and Emotional subscales. For Awareness, Distraction the internal and external reliability is just below the thresholds, placing the values in moderate reliability. For the External subscale, both values for internal and external validity are fairly valid. Although the limit for Cronbach is desirable to be above 0.7 as a rule of thumb, Taber discusses that there are instances where even lower values can be useful for validation [25].

For the construct validity, both types of professionals had overall scores indicating a more mindful approach to eating as compared to the individuals with excess weight. Except for the external subscale, for the other dimensions measured by the questionnaire, significant differences were recorded between people with excess weight and professionals (Table 3). Wansink [26], when discussing environmental factors such as the visibility, size, and accessibility of food, suggested that knowledge of the effect of these factors on food intake does not eliminate its impact, which could explain similar levels of external cues on “known groups”. For construct validity, others have used yoga professionals and recreational athletes and have obtained similar results [10,14]. Sports performance has been indirectly linked to mindfulness and stress reduction interventions [27]. In athletes, mindful training has improved coping skills, psychological well-being, and subjective sleep quality [28]. However, dietitians are trained and train others to change one’s overall approach to eating, including a transition from “mindless” eating and conscious eating, increasing awareness during eating for sustained lifestyle behavioral change [5]. Although other validations of the MEQ did not use dietitians as a part of the known group, our choice is backed up by the fact that behavioral training programs improve food choices beyond the nutrition knowledge, all these being mediators for mindful eating effects, as demonstrated by Kidwell [29].

The convergent validity explored the association between MEQ score and subscales scores with demographic variables such as gender, age, nutritional status and perceived levels of health status and stress (Table 4). Our results indicated that gender does not influence the overall mindfulness scores, comparable to Framson [10] and Kose [30]. Kose et al. [30] have reported similar non-significant differences for gender and have also found that mindful eating score had a direct relationship with age, meaning that as age increased, the mindfulness of eating improved, but with very small effect size, similar to our findings.

In this study, overweight and obesity were associated with more mindless eating and lower scores in all subscales (Table 4). A plethora of body of literature has been published on this subject [9,10,11,13,14,15,30]. It is well known that there is a high prevalence of eating disorders in individuals with excess weight; these conditions have a great potential to contribute to/and exacerbate each other [31,32]. The dysregulation model of obesity sustains that the inability to self-regulate eating behavior is due to the weak recognition of physical hunger and satiety cues [7]. Higher levels of emotional dysregulation and alexithymia were found in individuals with excess weight [33], and these feelings are an acquired trait due to inappropriate parenting practices [34].

Recently, higher levels of mindfulness were linked to better health status and lower levels of perceived stress [35,36,37]. Our results show that higher levels of perceived health status and lower levels of perceived stress were associated with higher levels of mindful eating, along with similar differences in subscales: Distraction, Disinhibition, Emotional and External subscale (Table 4). In a qualitative study, females identified stress as the primary trigger for emotional eating [38]. During the COVID-19 pandemic, several authors have linked psychological distress to emotional eating, both in adolescents [39] and adults [40].

In general, awareness is reached by being informed about a certain situation, gathering knowledge and understanding, and it applies both to internal and external factors. The awareness subscale is related to observing subtle details about the context and appearance of food. In the current validation, awareness was significantly higher in the dietitians (Table 3) and significantly lower in the individuals with excess weight and with lower self-perceived health status (Table 3 and Table 4).

Distraction is the disturbance of attention and can be an unhealthy mechanism, an avoidance strategy and an attentional disengagement strategy for decreasing negative emotions [41]. Chen et al. [42] have found that in daily life, distraction is more often used than problem solving and cognitive reappraisal because distraction requires fewer cognitive resources for modulation, and contrary to others [43], they consider that distraction is not avoidance but a ”temporary rest to strive for a better life”. In our sample, higher levels of distraction contributing to less mindful eating were observed in individuals with excess weight, those with higher levels of stress, and those with sub-optimal reported health status (Table 4).

According to Bryant [44], disinhibition is characterized by an inclination to overeat and eat opportunistically in response to negative affect, being unable to resist temptations. The greatest predictor of weight gain over 20 years was habitual disinhibition—overeating in response to cues from the environment, followed by emotional disinhibition—overeating as a response to emotions. Situational disinhibition—overeating on social occasions was not associated with weight gain [45]. In other research, disinhibition was positively correlated with perceived stress [46], poor diet quality, poorer health, and more problematic eating behaviors [47]. Our results are similar to published literature, showing higher disinhibition levels associated with excess weight, higher levels of stress and poorer health status (Table 4).

Emotional eating has been linked to emotional dysregulation, particularly to the inability to cope with negative emotions such as anxiety and irritability. The Emotionally Driven Eating Model states that due to the dysregulation of emotions, people exhibit maladaptive behavior based on eating in response to feelings [33]. In a large cross-sectional study [48], the highest scores associated with emotional eating were obtained by women compared to men and by former dieters, as compared to never dieters, and higher BMI as compared with lower BMI. Stress and unhealthy eating behavior, including high energy, fat, and sugar intake, increased number of meals, and fast-food frequency, were more prevalent in emotional eaters [49,50]. In our sample, emotional eating was associated with feminine gender, higher BMI, higher perceived levels of stress, and poor health status (Table 4).

Subjective senses of hunger and satiety are influenced by both internal cues from psychological processes and by external cues, which challenge the self-regulation of adequate food intake [51]. In a systematic review of functional neuroimaging studies [44], it was shown that females are more responsive to visual food stimuli compared with men. Belfort-DeAguilar et al. [52] have reviewed the role of food cues in the obesity pandemic and recognized the need to reduce the food cue reactivity as a separate point to touch during the weight loss approach. During stressful periods, there may be a shift of attention towards external cues, which could be the link between sensitivity to environmental factors associated with overeating [53]. Our results suggest that external cues influence at higher levels individuals with excess weight and individuals with higher levels of stress (Table 4).

This study has several limitations. Since the general populational sample was collected using social media, the generalizability of the results might be limited to individuals with access to technology. Additionally, selection bias might be an issue since individuals willing to participate in a survey about mindfulness of eating might be more interested in a healthier lifestyle. Although the validation of the whole questionnaire was moderate for the internal validation and with good reliability for the external validation, the validation of the five subscales was moderate for two subscales, acceptable only for two subscales and fair for one subscale. Therefore, some of the subscales are only moderately or fairly reliable. Other authors [15,18], validating the same questionnaire, have motivated that different questions might aggregate into different dimensions due to variations in the way questions are perceived and answered due to language and cross-cultural barriers. The seven-factor structure of the questionnaire specific to Romanian validation was not presented; instead, we chose to present the convergent validity, which is more generalizable and useful in public health interventions [54,55,56].

## 5. Conclusions

The Romanian version of the Mindful Eating Questionnaire is the first validated tool designed to investigate mindfulness of eating in Romanian adults, to the best of our knowledge. The questionnaire overall had adequate construct and convergent validity, moderate internal reliability and good external reliability, which make it a valuable tool both in clinical practice and research.

## Figures and Tables

**Figure 1 ijerph-19-10517-f001:**
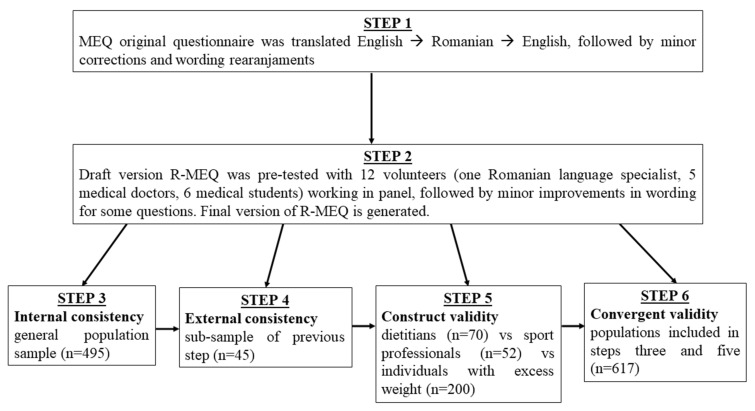
Flow diagram for the steps and populations used in reliability and validity testing of MEQ in a Romanian adult population.

**Table 1 ijerph-19-10517-t001:** Demographic variables per study group.

Demographic Variables		General Population (n = 495)	Dietitians (n = 70)	Sports Professionals (n = 52)	*p*-Value	Total
Feminine gender		81.2% (402) ^a^	87.1% (61) ^a^	23.1% (12) ^b^	<0.001	77.0% (475)
Excess weight		40.4% (200) ^a^	20.0% (14) ^b^	36.5% (19) ^a,b^	<0.001	37.9% (234)
In a relationship		78.0% (316) ^a^	78.6% (55) ^a^	71.2% (37) ^a^	0.521	77.4% (408)
Self-evaluation of health status as excellent or very good		35.6% (176) ^a^	62.9% (44) ^b^	67.3% (35) ^b^	<0.001	41.3% (255)
High levels of perceived stress		52.9% (262) ^a^	34.3% (24) ^b^	30.8% (16) ^b^	0.004	45.9% (242)
Categories BMI	Underweight and normal weight	59.6% (295)	80.0% (56)	63.5% (33)	<0.001	58.6% (309)
	Overweight	25.1% (124)	12.9% (9)	30.8% (16)		26.6% (140)
	Obese	15.4% (76)	7.1% (5)	5.8% (3)		14.8% (78)
Age (years) medians (IQR)		32.0 (20.0) ^a^	37.0 (16.0) ^b^	31.5 (13.0) ^a^	0.003	35.0 (17.0)
BMI (kg/m^2^) medians (IQR)		23.9 (6.7) ^a^	21.3 (4.4) ^b^	24.1 (4.4) ^a^	<0.001	23.7 (6.5)

Except for age and BMI, which are expressed as medians and (IQR), the numbers in this table represent percentages from the total and numbers in each category. To obtain the *p*-value, the Kruskal–Wallis test was applied. ^a, b^ Different superscript letter per row denotes a significant difference as compared to other groups using Mann–Whitney test with Bonferroni adjustment.

**Table 2 ijerph-19-10517-t002:** Cronbach’s alfa (n = 495), intraclass correlation coefficient and weighted kappa for the test-retest procedure (n = 45) in Romanian validation of MEQ.

Overall Questionnaire and Subscales	Cronbach’s Alfa (n = 495)	Intraclass Correlation Coefficient (n = 45)	Weighted Kappa (n = 45)
MEQ (28 items)	0.72	0.83	0.64
Awareness (7 items)	0.68	0.71	0.50
Distraction (8 items)	0.67	0.71	0.50
Disinhibition (3 items)	0.82	0.88	0.71
Emotional (4 items)	0.77	0.87	0.66
External (4 items)	0.55	0.60	0.45

**Table 3 ijerph-19-10517-t003:** MEQ score and subscales score per general population, dietitians and sport professionals.

Overall Questionnaire and Subscales	Individuals with Overweight and Obesity (n = 200)	Dietitians (n = 70)	Sports Professionals (n = 52)	*p*-Values of Kruskal–Wallis Test
MEQ total score	2.2 (0.5) ^a^	1.9 (0.6) ^b^	2.1 (0.8) ^b^	<0.001
Awareness	2.3 (0.7) ^a^	1.9 (0.6) ^b^	2.1 (0.7) ^a^	<0.001
Distraction	2.3 (1.0) ^a^	1.8 (0.7) ^b^	2.0 (0.7) ^b^	0.002
Disinhibition	2.1 (1.0) ^a^	1.6 (0.8) ^b^	1.7 (0.6) ^b^	<0.001
Emotional	2.0 (1.3) ^a^	1.5 (0.8) ^b^	1.3 (0.8) ^b^	<0.001
External	2.7 (0.7) ^a^	2.8 (0.7) ^a^	2.8 (0.8) ^a^	0.059

Values represent medians and interquartile range (IQR). ^a, b^ Different superscript letter per row denotes a significant difference as compared to other groups using Mann–Whitney test with Bonferroni adjustment.

**Table 4 ijerph-19-10517-t004:** MEQ score and subscales score per demographic variables (n = 617).

Variables and Categories		Awareness	Distraction	Disinhibition	Emotional	External	MEQ
Gender	Masculine (n = 142)	2.1 (0.9)	2.0 (0.7)	1.9 (0.9)	1.5 (1.3)	2.8 (0.8)	2.2 (0.4)
	Feminine (n = 475)	2.1 (0.7)	2.0 (1.0)	1.8 (0.9)	1.8 (1.3)	2.7 (0.7)	2.1 (0.4)
	*p*-value *	0.233	0.051	**0.002**	**<0.001**	0.100	0.452
Excess weight	no (n = 384)	2.1 (0.7)	2.0 (0.7)	1.6 (0.9)	1.8 (1.0)	2.8 (0.8)	2.1 (0.4)
	yes (n = 233)	2.3 (0.7)	2.3 (1.0)	2.1 (1.0)	1.9 (1.3)	2.7 (0.7)	2.2 (0.5)
	*p*-value *	**0.022**	**0.004**	**<0.001**	**<0.001**	**0.044**	**<0.001**
Perceived health status	Excellent or very good (n = 255)	2.1 (0.7)	2.0 (0.7)	1.6 (0.8)	1.5 (1.0)	2.8 (0.8)	2.1 (0.4)
	Good or less (n = 362)	2.1 (0.7) ***	2.3 (0.7)	1.9 (0.9)	1.8 (1.0)	2.7 (0.7)	2.2 (0.4)
	*p*-value *	**0.040**	**<0.001**	**0.001**	**<0.001**	0.198	**<0.001**
Stress	Low levels (n = 315)	2.1 (0.9)	2.0 (0.7)	1.8 (0.8)	1.5 (0.8)	2.8 (0.8)	2.1 (0.4)
	Higher levels (n = 302)	2.1 (0.9)	2.3 (1.0)	1.9 (1.0)	1.9 (1.3)	2.7 (0.7)	2.2 (0.4)
	*p*-value *	0.712	**<0.001**	**<0.001**	**<0.001**	**<0.001**	**<0.001**
Age (years)	Spearman’s rho	−0.011	−0.102	−0.147	−0.173	0.174	−0.147
	*p*-value **	0.786	**0.011**	**<0.001**	**<0.001**	**<0.001**	**<0.001**
BMI (kg/m^2^)	Spearman’s rho	0.128	0.085	0.311	0.187	−0.056	0.253
	*p*-value **	**0.001**	**0.034**	**<0.001**	**<0.001**	0.161	**<0.001**

For categorical variables, the values represent medians and interquartile range (IQR). For continuous variables, the values represent Spearman’s correlation coefficient. Significant *p*-values are marked in bold. * Mann–Whitney test ** Spearman’s test *** higher mean rank compared to excellent or very good category.

## Data Availability

The datasets used and/or analyzed during the current study are available from the corresponding author on reasonable request.

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
