# Peer review of "Mindful Eating Questionnaire: Validation and Reliability in Romanian Adults"

_ijerph, 2022, doi:10.3390/ijerph191710517_

Round 1
Reviewer 1 Report
In this manuscript, the authors tested the validity of the mindful eating questionnaire developed by Framson et al., for Romanian adult population. Overall, the manuscript is written well. The main concern that I have with this study is that the study does not include participitants from the general population that are not obese/overweight. It would be helpfule if the authors mentioned this fact as one of the limitations of the study in the discussion.
Author Response
In this manuscript, the authors tested the validity of the mindful eating questionnaire developed by Framson et al., for Romanian adult population. Overall, the manuscript is written well. The main concern that I have with this study is that the study does not include participitants from the general population that are not obese/overweight. It would be helpfule if the authors mentioned this fact as one of the limitations of the study in the discussion.
Dear reviewer,
Thank you for your comment. The general population sample includes 495 individuals, of which 200 are individuals with excess weight (overweight and obese). As presented now in Figure 1, the 495 individuals from the general population have been used to test internal consistency and convergent validity, but the 200 individuals with excess weight were used to test the construct validity against the dietitians and sports professionals.
Since we have participants that are not overweight and obese, and also individuals with overweight and obesity we consider that your request has already been fulfilled.

Reviewer 2 Report
In this manuscript, the authors aimed to determine the validity and reliability of a questionnaire to assess mindful eating among Romanian adults. Here are my comments:
[1] Introduction – it should be clearly stated that there is already a tool to assess mindful eating, but it has not been validated for Romanian adults (this is the objective of the current manuscript).
[2] Methods (line 77) – what does it mean by “pre-testing” in the step 2? Who were the 12 volunteers involved? Were they laypersons or qualified healthcare professionals? Was this “face validation”?
[3] Methods (lines 83-86) – should the reliability measure stated here refers to “internal consistency”?
[4] Methods (lines 95-99) – the “known-group” method was used for the construct validation. Firstly, it is essential to provide evidence from past studies that indicated that there is significant difference in mindful eating between dietitians/sport professionals and the general population so that this comparison can be used for the construct validation. Secondly, who were the sport professionals? Athletes? Please provide detailed information.
[5] Methods (lines 100-106) – the convergent validation should be conducted by Exploratory Factor Analysis using the Principal Component Analysis method, not just making comparisons the mindful eating score between several groups.
[6] Methods (lines 126-133) – how was the weight & height data collected considering an online questionnaire was administered? Self-reported?
[7] Results (Table 1) – What was the definition of “excess weight”?
[8] Results (Table 2) – The Cronbach’s alpha values for awareness, distraction, and external are below 0.7, with the lowest value being 0.55. This indicates poor internal consistency. Similarly, the ICC values for these subscales are below 0.75 (the preferred value). Weighted kappa should also be done and presented for the test-retest reliability.
[9] Results (Table 4) – It is odds that the awareness was significant different by the perceived health status despite the value was similar – 2.1 (0.7).
[10] Discussion (lines 225-227) – to indicate that the Cronbach’s alpha and ICC values are within the acceptable limits, it is necessary to provide references to support the statement, not just based on what values previously reported by other studies.
[11] Overall - the steps of validation can be presented in figure to clearly illustrate all the steps involved.
Author Response
Dear Reviewer,
Please find our answers to the issues raised by yourself in bold.
In this manuscript, the authors aimed to determine the validity and reliability of a questionnaire to assess mindful eating among Romanian adults. Here are my comments:
[1] Introduction – it should be clearly stated that there is already a tool to assess mindful eating, but it has not been validated for Romanian adults (this is the objective of the current manuscript).
Thank you for your comment. The introduction section has been updated in row 60
[2] Methods (line 77) – what does it mean by “pre-testing” in the step 2? Who were the 12 volunteers involved? Were they laypersons or qualified healthcare professionals? Was this “face validation”?
Thank you for your comment. The section has been updated in rows 77-78. This was not a face validation, because, as stated in the manuscript our purpose was to “review the understanding and improve the original meaning of each question”. Face validity is usually performed when a new instrument is introduced, which is not our case.
[3] Methods (lines 83-86) – should the reliability measure stated here refers to “internal consistency”?
Thank you for your comment. Text was updated with your suggestion.
[4] Methods (lines 95-99) – the “known-group” method was used for the construct validation. Firstly, it is essential to provide evidence from past studies that indicated that there is significant difference in mindful eating between dietitians/sport professionals and the general population so that this comparison can be used for the construct validation. Secondly, who were the sport professionals? Athletes? Please provide detailed information.
Thank you for your comment. The original article written by Framson [reference 10 Framson, C.; Kristal, A.R.; Schenk, J.M.; Littman, A.J.; Zeliadt, S.; Benitez, D. Development and Validation of the Mindful Eating Questionnaire. J. Am. Diet. Assoc. 2009, 109, 1439–1444, doi:10.1016/j.jada.2009.05.006.] used yoga practitioners and Pintado-Cucarella [reference 14 Pintado-Cucarella, S.; Rodríguez-Salgado, P. Mindful Eating and Its Relationship with Body Mass Index, Binge Eating, Anxiety and Negative Affect. J. Behav. Health Soc. Issues 2016, 8, 19–24, doi:10.1016/j.jbhsi.2016.11.003.] have used university students that practiced sports regularly, university athletes, and yoga practitioners. Since yoga is not so popular in Romania, we used sports professionals and our results were similar to references 10 and 14. There are no other studies showing directly a connection between dietitians and the mindfulness of eating, but the discussion section in 257-259 already contains an explanation of why the group of dietitians was chosen. Also, the section was updated rows 260-263 and a new reference was added [29. Kidwell, B.; Hasford, J.; Hardesty, D.M. Emotional Ability Training and Mindful Eating: J. Mark. Res. 2015, doi:10.1509/jmr.13.0188.]
[5] Methods (lines 100-106) – the convergent validation should be conducted by Exploratory Factor Analysis using the Principal Component Analysis method, not just making comparisons the mindful eating score between several groups.
Thank you for your comment. Several authors have failed to replicate the 5-factor structure described by Framson, motivating that different questions might aggregate into different dimensions due to variations in the way questions are perceived and answered due to language and cross-cultural barriers (reference 15 Abdul Basir, S.M.; Abdul Manaf, Z.; Ahmad, M.; Abdul Kadir, N.B.; Ismail, W.N.K.; Mat Ludin, A.F.; Shahar, S. Reliability and Validity of the Malay Mindful Eating Questionnaire (MEQ-M) among Overweight and Obese Adults. Int. J. Environ. Res. Public. Health 2021, 18, 1021, doi:10.3390/ijerph18031021.) and reference 18 Kömürcü Akik, B.; YiÄŸit, İ. Evaluating the Psychometric Properties of the Mindful Eating Questionnaire: Turkish Validity and Reliability Study. Curr. Psychol. 2022, doi:10.1007/s12144-021-02502-z.). Also, the structure of the sample population in other articles was different from our sample, mostly based general population from social networks. Framson’s sample, is mostly composed of individuals with high education/high social class (200 at one yoga studio; 100 at a university fitness facility; 40 at a weight loss program; 40 at a women’s weight loss and fitness facility; 40 at a software development company; 40 at a non-profit company; and 50 teachers and administrators at a preparatory school.) and the sample gathered by Abdul Basir includes 144 overweight and obese working adults conveniently recruited in a selected public university. Our validation did not produce a 5-factor structure similar to Fromson’s, but a 7-factor structure (data not shown). Instead of presenting the structure specific only to the Romanian population, we decided to present it as a convergent validation using the anthropo-demographic variables, which could be of general importance in future public health-related interventions, similar to other validation that our group had done (https://www.nature.com/articles/s41430-020-0616-5). Since the 7-factor structure specific to Romanian validation was not presented, it was added as a limitation of this manuscript rows 339-341.
[6] Methods (lines 126-133) – how was the weight & height data collected considering an online questionnaire was administered? Self-reported?
Thank you for your comment. The section was updated - row 112
[7] Results (Table 1) – What was the definition of “excess weight”?
Thank you for your comment. A definition was added – rows 134-136
[8] Results (Table 2) – The Cronbach’s alpha values for awareness, distraction, and external are below 0.7, with the lowest value being 0.55. This indicates poor internal consistency. Similarly, the ICC values for these subscales are below 0.75 (the preferred value). Weighted kappa should also be done and presented for the test-retest reliability.
Thank you for your comment. We have added the interpretation of Cronbach’s alpha and ICC with references in rows 94-95, and the weighted kappa – rows 140-141, and as requested in Table 2.
[9] Results (Table 4) – It is odds that the awareness was significant different by the perceived health status despite the value was similar – 2.1 (0.7).
Thank you for your comment. Values in the table represent medians and IQR, as explained in the table footer. The comparative test (Mann-Whitney test) is not comparing directly the medians but is using the ranking of data, therefore these types of discrepancies could be found. An explanation was added in the footer in Table 4 to explain this issue.
[10] Discussion (lines 225-227) – to indicate that the Cronbach’s alpha and ICC values are within the acceptable limits, it is necessary to provide references to support the statement, not just based on what values previously reported by other studies.
Thank you for your comment. Ranges and references were added in the method section- rows 141-147 and 243-245 with references
[11] Overall - the steps of validation can be presented in figure to clearly illustrate all the steps involved.
Thank you for your comment. Figure 1 was added to the manuscript.

Reviewer 3 Report
The effective use of the questionnaires available in the literature enables the assessment of their application in appropriate reference groups under specific regional conditions. Therefore, it seems advisable to clarify: what were the inclusion criteria for people from the general population?
In spite of the lack of influence of gender on the examined parameters, could a significant share of women in the group of people participating in the study have influenced the results of the study?
In discussing the results of the study, it would be helpful to relate them to the diet of people participating in it. Do the authors of the study have information on the eating behavior of the study participants that could be confronted with the results of the questionnaire?
How could be explain the average or small size effect of the MEQ score comparison effect between overweight and obese people and nutritionists and athletes? Could this affect the conclusions of the presented work?
Author Response
Dear Reviewer,
Please find our answers to the issues raised by yourself in bold.
The effective use of the questionnaires available in the literature enables the assessment of their application in appropriate reference groups under specific regional conditions. Therefore, it seems advisable to clarify: what were the inclusion criteria for people from the general population?
Thank you for your comment. Since our target was the general population and the sample was recruited mostly using social media, we did not have any restrictions for inclusion. Since all questions were required for the completion of the questionnaire, only completed questionnaires were received through the platform. These aspects were added in the methodology section rows 117-119.
The risk of recruitment bias is mentioned in the limitations of this study.
In spite of the lack of influence of gender on the examined parameters, could a significant share of women in the group of people participating in the study have influenced the results of the study?
Thank you for your comment. Mindfulness is not gender-specific, as others (Framson et al -reference 10 and Köse et al reference 30) have also found, which is already discussed. Sex differences are reported though in other components of mindful eating as already presented in Table 4: women have higher scores in emotional eating subscale and men have higher scores in disinhibition subscale). So, the populational structure did not influence the results.
In discussing the results of the study, it would be helpful to relate them to the diet of people participating in it. Do the authors of the study have information on the eating behavior of the study participants that could be confronted with the results of the questionnaire?
Thank you for your comment. Since the purpose of this manuscript is to validate the psychometric properties of the questionnaire and to validate it in the Romanian language, the evaluation of the food intake was not included in this project. But it might be an excellent idea to use the questionnaire, after the publication of the validation.
How could be explain the average or small size effect of the MEQ score comparison effect between overweight and obese people and nutritionists and athletes? Could this affect the conclusions of the presented work?
Thank you for your comment. As presented in rows 198-202, the effect size of the comparison between individuals with overweight and obesity and dietitians was medium and between individuals with overweight and obesity and sports professionals was small to medium. There are several authors (Wansink – reference 26 and Kidwell -reference 29), as already presented in the discussion section that have underlined that the knowledge of nutrition is not sufficient to inhibit the influence of external factors on temptation and food intake. Therefore, we consider these differences as adequate for our research hypothesis, that individuals with excess weight, in general consume less mindful than dietitians and sports professionals.

Round 2
Reviewer 2 Report
The authors have made the required revisions in this revised manuscript. However, there are a few major comments:
[1] As stated at line 94, the acceptable Cronbach's alpha value should be > 0.7, but three components (i.e., awareness, distraction, and external) had Cronbach's alpha values below 0.7, which is the acceptable value. Can the internal consistency of this questionnaire be assumed?
[2] The acceptable weighted Kappa value is at least 0.60 to indicate good agreement. Similarly, the weighted Kappa values for three components were all below 0.60.
[3] The preferred ICC value should be > 0.75 to indicate good reliability. However, the authors chose a much lower cutoff, 0.50, which only indicates moderate reliability.
Therefore, if the values obtained from the analyses are not meeting the acceptable cutoff, it is important to rectify the issues and re-run the analyses. If the authors would like to proceed with the current data, then please clearly state that it only has "moderate" internal consistency or reliability so that the researchers who wish to use this instrument will be aware of the limitations.
[4] Convergent validity refers to how closely the new scale is related to other variables and other measures of the same construct. Please provide a reference (beside the authors' own publication) that using the anthropo-demographic variables is a valid method to test for convergent validity.
